# Research on the Prediction Model of Engine Output Torque and Real-Time Estimation of the Road Rolling Resistance Coefficient in Tracked Vehicles

**DOI:** 10.3390/s23177549

**Published:** 2023-08-31

**Authors:** Weijian Jia, Xixia Liu, Guodong Jia, Chuanqing Zhang, Bin Sun

**Affiliations:** 1Army Academy of Armored Forces, Beijing 100072, China; jwjyxsl@163.com (W.J.);; 2CITIC Machinery Manufacturing Inc., Linfen 043000, China

**Keywords:** tracked vehicles, engine output torque prediction model, GA–BP neural network, estimation of rolling resistance coefficient

## Abstract

Road parameter identification is of great significance for the active safety control of tracked vehicles and the improvement of vehicle driving safety. In this study, a method for establishing a prediction model of the engine output torques in tracked vehicles based on vehicle driving data was proposed, and the road rolling resistance coefficient f was further estimated using the model. First, the driving data from the tracked vehicle were collected and then screened by setting the driving conditions of the tracked vehicle. Then, the mapping relationship between the engine torque $T_{e}$, the engine speed $n_{e}$, and the accelerator pedal position β was obtained by a genetic algorithm–backpropagation (GA–BP) neural network algorithm, and an engine output torque prediction model was established. Finally, based on the vehicle longitudinal dynamics model, the recursive least squares (RLS) algorithm was used to estimate the f. The experimental results showed that when the driving state of the tracked vehicle satisfied the set driving conditions, the engine output torque prediction model could predict the engine output torque ${\hat{T}}_{e}$ in real time based on the changes in the $n_{e}$ and β, and then the RLS algorithm was used to estimate the road rolling resistance coefficient $\hat{f}$. The average coefficient of determination R of the ${\hat{T}}_{e}$ was 0.91, and the estimation accuracy of the $\hat{f}$ was 98.421%. This method could adequately meet the requirements for engine output torque prediction and real-time estimation of the road rolling resistance coefficient during tracked vehicle driving.

## 1. Introduction

Tracked vehicles are mostly used in agricultural, fire protection, and military fields due to their good trafficability. A driving road is complex and changeable, and the demands on the dynamics and safety of tracked vehicles are high [1]. The road parameters affect the acceleration, braking, and steering performance of vehicles, and they are important parameters for risk assessment and active safety control during tracked vehicle driving [2]. Road parameter estimation is of great significance for improving the safety and dynamic performance of vehicles [3]. Road identification is generally achieved by estimating a parameter that reflects the characteristics of the road surface, such as the road adhesion coefficient, roughness, rolling resistance coefficient, or slope. To realize road surface recognition and improve the safety of vehicle driving, researchers have conducted in-depth studies. For tracked vehicles, researchers have mostly focused on the dynamic characteristics of tracked vehicles [4] and the coupling between the track and ground during vehicle driving. There have been few studies on the further application of research results to road recognition.

At present, there are two main methods to realize road recognition. One is to directly measure the road through environmental sensors or vehicle state sensors [5,6]. Abhinav et al. [7] proposed a terrain recognition method based on a deep learning long short-term memory model, using the acoustic waves generated by the interactions between the vehicle and the terrain as the terrain feature variable. Huang et al. [8] proposed a road boundary monitoring method based on a deep learning network. The experimental results showed that the method had high accuracy and robustness for lane boundary line monitoring in various scenarios. Based on road images, laser radar point cloud data, and vehicle state information, Zhao et al. [9] realized road recognition by parameter estimation and state fusion. The direct measurement method using environmental sensors for road recognition has the advantages of real-time measurement capabilities and high estimation accuracy. However, to achieve large-scale commercial applications, it is necessary to further reduce the cost of environmental sensors and their reliability in harsh environments [10].

Another method is to estimate the road feature parameters through a vehicle model and then perform road recognition. The method of estimating the road adhesion coefficients of wheeled vehicles based on the slip-slope method [11] using the μ−s curve is comparatively mature, but the implementation of this method is based on an accurate tire model. Since the vehicle longitudinal dynamics model does not include the tire model, the method of estimating the road characteristics parameters based on this model is theoretically more suitable for tracked vehicles. The application of this method to wheeled vehicles could also be used as a reference. The estimation of the engine output torque is a difficult task for road parameter estimation based on the vehicle longitudinal dynamics model. Chu et al. [12] used the vehicle longitudinal dynamics model to estimate the road slope by making full use of the accurate driving force information from an electric-drive vehicle. Liu et al. [13] established an engine output torque prediction model, including a fuel supply system model, an in-cylinder combustion model, and a crankshaft dynamics model. Based on the vehicle longitudinal kinematics model, the Kalman filter algorithm was used to estimate the road slope. Because the established engine output torque prediction model was too complex, this method was difficult to use for real vehicle control. Cong et al. [14] established a look-up table model of the engine output torque by fitting the corresponding relationship between the engine output torque, engine speed, and accelerator pedal position and estimated the road slope based on the Kalman filter algorithm. This method required a lot of manual calibration work in the bench test stage of the vehicle, and it was difficult to update after the calibration was complete. The predicted torque error of the model would increase with the engine performance degradation. The development of intelligent algorithms provides a new idea for establishing an engine model with nonlinearity, multiple disturbances, and time lag [15].

In this study, the tracked vehicle was the research object, and a method for establishing a prediction model of the engine output torque in the tracked vehicle, based on the vehicle driving data, was proposed. This method saves a lot of manual calibration work during the vehicle bench test stage and has the advantage of enabling real-time updates. The rolling resistance coefficient estimation for the road was realized using the model. The vehicle driving data, during the driving process, from the tracked vehicle were collected. The vehicle driving data were screened by setting the driving conditions of the vehicle to estimate the f. The Kalman filter algorithm was used to filter the longitudinal acceleration a in the selected data segments, and the engine output torque $T_{e}$ of each data segment was calculated based on the filtered longitudinal acceleration $\hat{a}$. The engine speed $n_{e}$, the engine speed variation rate $n_{e}^{\prime }$, the accelerator pedal position β, and the accelerator pedal position variation rate $\beta^{\prime }$ were the inputs, and the engine output torque $T_{e}$ was the output. A genetic algorithm–backpropagation (GA–BP) neural network algorithm was used to fit the mapping relationship between the inputs and the output, $T_{e}=f(n_{e},n_{e}^{\prime },\beta ,\beta^{\prime })$. Based on the longitudinal dynamics model of tracked vehicles, the RLS algorithm with the forgetting factor λ was used to estimate the $\hat{f}$. The implementation process is shown in Figure 1. The experimental results showed that the sensor information could be used to automatically judge the driving conditions during the driving process of the tracked vehicle. When the driving conditions of the vehicle satisfied the set driving conditions, the engine output torque prediction model predicted the ${\hat{T}}_{e}$ in realtime based on the $n_{e}$ and β, and then it further estimated the $\hat{f}$. The estimation results had high accuracy and could better meet the requirements for the real-time estimation of the road parameters for tracked vehicles.

## 2. Estimation of f Based on Recursive Least Squares (RLS) Algorithm

The longitudinal driving forces on the tracked vehicle are shown in Figure 2. The vehicle driving force $F_{t}$ can be expressed as follows:(1)$$F_{t}=mgf\cos\alpha +\delta ma+mg\sin\alpha +\frac{C_{D}A}{21.15}v^{2},$$
where m is the mass of the vehicle, g is the acceleration of gravity, f is the rolling resistance coefficient of the road, α is the road slope, δ is the rotating mass scaling factor, a is the longitudinal acceleration of the tracked vehicle, $C_{D}$ is the air resistance coefficient, A is the windward area of the tracked vehicle, and v is the speed of the tracked vehicle.

The relationship between the $T_{e}$ and vehicle driving force $F_{t}$ can be expressed as follows:(2)$$T_{e}=\frac{F_{t}r}{i\eta },$$
where i is the transmission ratio from the engine to the driving wheel, η is the transmission efficiency, and r is the radius of the sprocket.

The formula for f can be obtained by combining (1) and (2):(3)$$f=\frac{\frac{T_{e}i\eta }{r}-\frac{C_{D}A}{21.15}v^{2}-mg\sin\alpha -\delta ma}{mg\cos\alpha }.$$

From Formula (3), it can be seen that f can be obtained when $T_{e}$, a, and α are known.

The RLS algorithm with the forgetting factor λ is used to estimate the f. The RLS algorithm can be expressed as follows:(4)$$y(t)=\varphi^{T}(t)\theta (t)+e(t),$$
where φ(t) is the estimated parameter vector at time t, θ(t) is the regression vector at time t, and e(t) is the deviation between the measured value y(t) and the estimated value $\varphi^{T}(t)\theta (t)$ at time t.

The RLS algorithm iteratively updates the position parameter vector φ(t) at each sampling time by making the regression vector θ(t) contain the input and output data from a previous time. The RLS algorithm minimizes the estimation bias for each iteration period by updating the vector regression θ(t). In this paper, $y(t)=T_{e}i\eta /r-\frac{C_{D}A}{21.15}v^{2}-mg\sin\alpha (t)-\delta ma(t)$, θ(t)=mgcosα(t), and λ=0.98.

The calculation steps for the RLS algorithm at each time t are as follows:
(1)The system output y(t) is measured, and the regression vector θ(t) is calculated.(2)The difference e(t) between the actual output of the system y(t) at time t and the output of the prediction model obtained by estimating the parameters $\varphi^{T}(t)\theta (t-∆t)$ is calculated. ∆t is the time interval. The difference e(t) can be expressed as follows:(5)$$e(t)=y(t)-\varphi^{T}(t)\theta (t-∆t).$$(3)The updated gain vector G(t) and the covariance matrix C(t) are calculated. These can be expressed as follows:(6)$$C(t)=\frac{1}{\lambda }\left[C(t-∆t)-\frac{C(t-∆t)\varphi (t)\varphi^{T}(t)C(t-∆t)}{\lambda +\varphi^{T}(t)P(t-∆t)\varphi (t)}\right],$$
(7)$$G(t)=\frac{C(t-∆t)\varphi (t)}{\lambda +\varphi^{T}(t)C(t-∆t)\varphi (t)}.$$(4)The parameter estimation vector φ(t) is updated as follows:(8)φ(t)=φ(t−∆t)+Ge(t).

## 3. Tracked Vehicle Driving Data Acquisition and Processing

The tracked vehicle examined in this study was equipped with a diesel engine, a dry clutch, and a fixed shaft gearbox. The tracked vehicle was equipped with a combined inertial navigation module, including an acceleration sensor and a gyroscope. The positioning data were processed by differential processing, and the accuracy reached the centimeter level. The acceleration sensor was used to measure the longitudinal acceleration value of the tracked vehicle in real time. The gyroscope was used to measure the vehicle pitch angle, and the vehicle pitch angle was assumed to be equal to the road slope value. The vehicle controller received the driver’s control instructions to control the vehicle, and the driving data recorder recorded the vehicle’s driving data, which was convenient for the data analysis and control optimization of the vehicle. The communication structure of each module is shown in Figure 3. 

The test site was a vehicle driving test site in Shanxi, China. The total length of the test site route was 10 km, including a sand road and a cement road. The rolling resistance coefficients of the two roads were measured to be 0.06 and 0.045. Figure 4 shows the satellite map of the experimental site. A total of 119.67 h driving data were collected by the driving data recorder. The collected data included GPS coordinates, vehicle speed, longitudinal acceleration, pitch angle, heading angle, and gear and clutch displacement.

To ensure the accuracy of the engine output torque model, the driving data were screened by setting the driving conditions for the tracked vehicle. The vehicle driving data that satisfied the driving conditions were used as the effective data to establish the engine output torque prediction model.

(1)The measurement of the vehicle pitch angle by the gyroscope was affected not only by the road slope but also by installation error, the suspension state, and other factors. Under some conditions, for example, the clutch was engaged too fast when shifting, which caused the vehicle to pitch in a short time even if it was driving on a flat road, resulting in measurement errors. At the same time, a large change in the road slope also increased the measurement error of the acceleration sensor. To improve the prediction accuracy of the model, the driving data with a large angle measured by the gyroscope were eliminated by setting the ramp threshold to $\alpha_{th}=3{}^{\circ}$, so that the tracked vehicle could drive on a flat road, which was approximately level, as far as possible.(2)The selected vehicle driving data did not include the clutch separation process, and the driving force during the vehicle driving process was only provided by the engine. The engagement and separation state of the clutch was judged by the displacement of the clutch control cylinder ${clh}_{x}$. When the clutch combination displacement ${clh}_{x}\leq 18\;\mathrm{mm}$, the clutch was considered engaged. Setting the vehicle acceleration threshold $a_{th}$ and the minimum stable driving time threshold $t_{s}$ ensured that the stable driving data were screened after the vehicle shift was complete. By analyzing the driving data, we set $a_{th}=0.4\;m/s^{2}$ and $t_{s}=10\;s$. The driving data when the vehicle acceleration $\left|a\right|\leq 0.4\;m/s^{2}$ for more than 10 s after the clutch was engaged was considered stable and valid data.(3)It was necessary to limit the heading angle γ in the screened vehicle driving data to ensure that the tracked vehicle was in a straight-line driving state. Considering the influence of sensor measurement error and random road disturbances, we set $\gamma_{th}=5{}^{\circ}$. In the selected driving data, the change in the heading angle of the vehicle between the initial moment and the final moment could not exceed 5°.

Due to the body vibration and acceleration sensor measurement bias during the running of the tracked vehicle, the acceleration measurement data had a larger error than the real value. The Kalman filter algorithm was used to filter a to obtain an accurate longitudinal acceleration $\hat{a}$. The vehicle displacement p, velocity v, and acceleration a were the state variables, and u was the measured value of the acceleration sensor. The driving displacement $p_{GPS}$ and the driving speed $v_{GPS}$ obtained by the vehicle’s integrated inertial navigation module through the differential positioning system were taken as the observed quantities. The vehicle state equation at time t can be expressed as follows:(9)$$\left[\begin{array}{c} p_{t}\\ v_{t}\\ a_{t} \end{array}\right]=\left[\begin{array}{ccc} 1 & ∆t & \frac{1}{2}∆t^{2}\\ 0 & 1 & ∆t\\ 0 & 0 & 0 \end{array}\right]\left[\begin{array}{c} p_{t-∆t}\\ v_{t-∆t}\\ a_{t-∆t} \end{array}\right]+\left[\begin{array}{c} 0\\ 0\\ 1 \end{array}\right]u_{t-∆t},$$
(10)$$\left[\begin{array}{c} p_{{GPS}_{t-∆t}}\\ v_{{GPS}_{t-∆t}}\\ \frac{v_{{GPS}_{t}}-v_{{GPS}_{t-∆t}}}{∆t} \end{array}\right]=\left[\begin{array}{ccc} 1 & 0 & 0\\ 0 & 1 & 0\\ 0 & 0 & 1 \end{array}\right]\left[\begin{array}{c} p_{t-∆t}\\ v_{t-∆t}\\ a_{t-∆t} \end{array}\right],$$
where $X_{k}=\left[\begin{array}{c} p_{t}\\ v_{t}\\ a_{t} \end{array}\right]$, $Y_{k}=\left[\begin{array}{c} p_{{GPS}_{t-∆t}}\\ v_{{GPS}_{t-∆t}}\\ \frac{v_{{GPS}_{t}}-v_{{GPS}_{t-∆t}}}{∆t} \end{array}\right]$, $F=\left[\begin{array}{ccc} 1 & ∆t & \frac{1}{2}∆t^{2}\\ 0 & 1 & ∆t\\ 0 & 0 & 0 \end{array}\right]$, $B=\left[\begin{array}{c} 0\\ 0\\ 1 \end{array}\right]$, $H=\left[\begin{array}{ccc} 1 & 0 & 0\\ 0 & 1 & 0\\ 0 & 0 & 1 \end{array}\right]$, and ∆t is the calculation time step.

The Kalman filter algorithm uses a recursive method to solve the filtering problem of discrete linear data [16]. The steps are as follows:(1)Update the prediction equation:(11)$$\left\{\begin{array}{c} {\bar{X}}_{t}^{-}=F{\bar{X}}_{t-∆t}+Bu_{t}\\ P_{t}^{-}=FP_{t-∆t}F^{T}+Q \end{array}\right..$$(2)Update the Kalman gain coefficient:(12)$$K=P_{t}^{-}H^{T}{\left(HP^{-}H^{T}+R\right)}^{-1}.$$(3)Update the measurement equation:(13)$$\left\{\begin{array}{c} {\bar{X}}_{t}={\bar{X}}_{t}^{-}+K(Y_{t}-H{\bar{X}}_{t}^{-})\\ P_{t}=(I-KH)P_{t}^{-} \end{array}\right..$$

In these formulas, ${\bar{X}}_{t}^{-}$ is the prior state estimation at time t, ${\bar{X}}_{t}$ is the posterior state estimation at time t, $P_{t}^{-}$ is the prior covariance matrix at time t, $P_{t}$ is the covariance matrix at time t, Q is the process noise covariance matrix, R is the observation noise covariance matrix, and K is the Kalman gain coefficient. In this study, $Q=\left[\begin{array}{ccc} 0.5 & 0 & 0\\ 0 & 0.5 & 0\\ 0 & 0 & 0.5 \end{array}\right]$, $R=\left[\begin{array}{ccc} 0.2 & 0 & 0\\ 0 & 0.2 & 0\\ 0 & 0 & 0.2 \end{array}\right]$, and $P_{0}=\left[\begin{array}{ccc} 0.1 & 0 & 0\\ 0 & 0.1 & 0\\ 0 & 0 & 0.1 \end{array}\right]$. The acceleration data measured by some accelerometers are filtered, and the filtering effect is shown in Figure 5.

## 4. Engine Output Torque Prediction Model

After obtaining $\hat{a}$ using the Kalman filter algorithm, the driving force $F_{t}$ of the vehicle was calculated according to Formula (1), and the $T_{e}$ was further obtained. The $n_{e}$, $n_{e}^{\prime }$, β, and $\beta^{\prime }$ were used as the inputs in the engine output torque prediction model, and the corresponding engine output torque $T_{e}$, calculated by $\hat{a}$, was used as the model output. The GA–BP neural network algorithm was trained on the inputs and output, and the mapping relationship between the $T_{e}$ and $n_{e}$, $n_{e}^{\prime }$, β, and $\beta^{\prime }$ was $T_{e}=f(n_{e},n_{e}^{\prime },\beta ,\beta^{\prime })$. Thus, the engine output torque prediction model was established.

The GA–BP neural network algorithm makes use of the global optimization ability of the genetic algorithm to make up for the shortcomings of BP neural networks, such as the slow learning convergence speeds, uncertain network structures, and ease of falling into the local minimum. The initial weights and thresholds in the BP neural network were used as genes in the genetic algorithm. The values on the genes represented the connection weights or thresholds in the BP neural network and formed the chromosomes of the genetic algorithm. A certain number of chromosomes were used as the initial population of the genetic algorithm. After selection, crossover, and mutation iterations, the initial weights and thresholds of the optimal BP neural network were obtained. The GA–BP neural network algorithm structure diagram is shown in Figure 6. After the simulation test, the number of genetic iterations in the genetic algorithm is set to 30, the number of populations is 5, the probability of crossover is 0.7, and the probability of mutation is 0.1.

The BP neural network continuously corrected the weights and thresholds of each neural network layer through error backpropagation. The number of input nodes in the BP neural network is four, the number of output nodes is one, and the hidden layer is ten. When the training results met the set requirements, the training was stopped and the prediction results were output. The network structure is shown in Figure 7.

In the graph, $\omega_{jk}^{\left[n\right]}$ is the weight value from the *k*-th node to the *j*-th node of the (n−1)-th layer in the neural network, $b_{j}^{\left[n\right]}$ represents the threshold of the *j*-th node of the n-th layer neural network, $z_{j}^{\left[n\right]}$ is the linear result of the *j*-th node added to the *n*-th layer neural network, and $a_{j}^{\left[n\right]}$ represents the output value of the *j*-th node of the *n*-th layer neural network. 

The initial weights and thresholds for each layer of the neural network were calculated by the genetic algorithm to obtain the optimal solution. σ denotes the activation function, and χ donates the learning rate. The input signal fitting process is as follows:

Gradient of output layer:(14)$$\sigma_{j}^{\left[n\right]}=\frac{\partial s}{\partial \alpha_{j}^{\left[n\right]}}\sigma^{\prime }\left(z_{j}^{\left[s\right]}\right);$$

Gradient of hidden layer:(15)$$\sigma_{j}^{\left[n\right]}=\sum \omega_{kj}^{\left[n\right]}\sigma_{k}^{\left[n+1\right]}\sigma^{\prime }\left(z_{j}^{\left[n\right]}\right);$$

*s*-th iteration threshold:(16)$$b_{j}^{\left[n\right]}(s)=b_{j}^{\left[n\right]}(s-1)-\eta \sigma_{j}^{\left[n\right]};$$

*s*-th iteration weight:(17)$$\omega_{jk}^{\left[n\right]}(s)=\omega_{jk}^{\left[n\right]}(s-1)-\eta \sigma_{j}^{\left[n\right]}a_{k}^{\left[n-1\right]}.$$

The learning rate χ was adaptively adjusted according to the error change e(s), which can be expressed as:(18)$$\chi (s)=\left\{\begin{array}{cc} 1.05\chi (s-1) & e(s-1)<e(s-2)\\ 0.5\chi (s-1) & e(s-1)<1.04e(s-2)\\ \chi (s-1) & other \end{array}\right..$$

## 5. Experimental Results and Analysis

The accuracy of the tracked vehicle engine output torque prediction model and the effectiveness of the f estimation method were verified by experiments. The experimental pavement was a sand road and a cement road. The structural parameters of the experimental vehicle are shown in Table 1.

To improve the computational efficiency, the engine output torque prediction model was established through offline updates and online prediction. In this study, MATLAB 2021b was used to train the engine output torque prediction model offline through the selected vehicle driving data, and the generated model was converted into C code and imported into an industrial personal computer (IPC). 

The IPC received the vehicle state in real time through the controller area networks (CAN) bus. When the vehicle state was determined to meet the set working conditions, a data storage container was established to store the vehicle state data. When it was determined that the current vehicle driving state did not meet the set conditions, the data stored in the container were emptied and the vehicle state was continuously monitored. When the container stored data for more than 10 s, i.e., the vehicle had been running in a specific state for 10 s, the engine output torque prediction model began to predict the ${\hat{T}}_{e}$ based on the $n_{e}$ and β, and the $\hat{f}$ was further estimated. The data container was used to store data to estimate the $\hat{f}$ and update the $\hat{f}$ in real time, based on the current vehicle state. The process through which the IPC processed the tracked vehicle driving data is shown in Figure 8.

### 5.1. Estimation of $\hat{f}$ for Tracked Vehicles Driving on a Sand Road

The annular sand road in the vehicle driving test field was selected as the experimental test road. According to the driving habits and environmental conditions, the driver determined the gear and speed to maintain driving safety and speed. The IPC received the driver’s control information and the vehicle driving state data in real time, and automatically determined whether the vehicle’s state satisfied the set driving conditions. When the driving conditions were satisfactory, the engine output torque prediction model predicted the ${\hat{T}}_{e}$ in real time and estimated the $\hat{f}$.

Figure 9 shows the trajectory of the tracked vehicle. The vehicle started from the starting point and traveled around the circular runway. The driving distance was 3694 m. The red solid line section marked in Figure 9 indicates that the state of the tracked vehicle on this section satisfied the set driving conditions. The total length of the red solid line segment was 2054.5 m. Figure 10 shows the change in the speed, and the gear and clutch cylinder displacement in the tracked vehicle during the whole driving process. As can be seen from Figure 10, the total driving time of the vehicle was 622.3 s. The vehicle starts in second gear, the highest gear was fifth gear, the highest speed was 32.5 km/h, the commonly used gear during the vehicle driving was fourth gear, and the average driving speed was 20.55 km/h. When ${flag}_{state}=1$, the driving state of the vehicle meets the set conditions. The entire driving process satisfied the set driving conditions during 10 periods, and the total time was 313.8 s. Figure 11 shows the changes in the acceleration, pitch angle, and heading angle of the tracked vehicle. It can be seen from Figure 11 that the acceleration of the vehicle increased rapidly in a short time during the engagement of the vehicle’s clutch, and the impact of shifting the vehicle was large. After the clutch engagement was completed, the vehicle acceleration changed relatively smoothly when the vehicle was accelerating and decelerating. The acceleration measurement value was more credible, and it was reasonable to screen the vehicle driving data by setting the acceleration threshold. It can be seen from the change in the heading angle that when the tracked vehicle was under the set driving conditions, the heading angle of the vehicle was almost unchanged, and the vehicle could be considered to have maintained, approximately, a straight driving state. Under the set driving conditions, the change in the pitch angle of the tracked vehicle was in the range of the set pitch angle. Figure 12 shows the changes in the engine speed and accelerator pedal angle during the driving process of the vehicle. The selected driving information data excluded the rapid variation stage of the engine speed during the shifting process. The accelerator pedal angle and engine speed varied smoothly, and the vehicle ran stably.

Figure 13 shows the ${\hat{T}}_{e}$ prediction of the engine output torque prediction model when the tracked vehicle satisfied the driving conditions for the first time on the sand road. The $T_{e}$, calculated based on $\hat{a}$ using Equation (1), was the real value, and the engine output torque predictions of the BP neural network were used as the control data. The engine output torque value ${\hat{T}}_{e_{GA-BP}}$ predicted by the GA–BP neural network was closer to the $T_{e}$. The root mean square error $\sigma_{e}$ and the coefficient of determination R were used as the evaluation indices on the accuracy of the engine output torque estimation.
(19)$$\sigma_{e}=\sqrt{{\left({\hat{T}}_{e}-T_{e}\right)}^{2}/n},$$
(20)$$R=1-\frac{\sum {\left(T_{e}-{\hat{T}}_{e}\right)}^{2}}{\sum {\left(T_{e}-{\bar{T}}_{e}\right)}^{2}},$$
where n is the number of sample data, and ${\bar{T}}_{e}$ is the average value of the true values on the output torque of the transmitter calculated from the sample data.

The root mean square error of the engine output torque obtained by the engine output torque prediction model established by the BP neural network was 78.65, and the coefficient of determination was 0.768. The root mean square error of the engine output torque calculated by the GA–BP was 45.06, and the coefficient of determination was 0.924, which was 42.71% and 20.31% higher than those of the BP neural network, respectively. Figure 14 shows the result on the further f estimation by the RLS algorithm. The rolling resistance coefficient estimated by the GA–BP neural network method was more convergent. The average error of the $\hat{f}\;$ value estimated by the RLS algorithm was 0.00093, while it was 0.00117 for the BP neural network. Thus, the estimation accuracy was improved by 20.51%.

Table 2 shows the vehicle state and ${\hat{T}}_{e}$ prediction for the 10 periods when the tracked vehicle satisfied the driving conditions. The average root mean square error of the engine output torque estimated by the GA–BP neural network was 42.24, and the average root mean square error calculated by the BP neural network was 73.95. The average root mean square error of the engine output torque calculated by the GA–BP neural network on the sand road was 42.87% higher than that of the BP neural network. Similarly, the average coefficient of determination of the engine output torque estimated by the engine output torque model established by the GA–BP neural network was 0.918, which was 27.73% higher than that of the BP neural network.

Figure 15 shows the change in the average absolute error $e_{mb}$ of the $\hat{f}$ estimation results obtained by the two methods, which was calculated as
(21)$$e_{mb}=\frac{\sum \left|{\hat{T}}_{e}-T_{e}\right|}{n}.$$

The $e_{mb}$ value of the road rolling resistance coefficient estimated by the GA–BP neural network was 0.00108, which was 24.44% higher than that of the BP neural network. The engine output torque model was established by the GA–BP neural network on the sand road, and the engine output torque was estimated. The estimation accuracy of the rolling resistance coefficient of the road was significantly improved compared with that of the BP neural network. Thus, the GA–BP neural network can better meet the real-time estimation requirements of the rolling resistance coefficient during the driving process of a vehicle on a sand road.

### 5.2. Estimation of $\hat{f}$ for Tracked Vehicles Running on a Cement Road

The driver drove the tracked vehicle by starting in second gear on the cement road. During the driving process of the vehicle, the IPC received the state information on the vehicle, predicted the engine output torque, and estimated the road rolling resistance coefficient in real time, when the vehicle state satisfied the driving conditions. 

Figure 16 shows the driving route of the tracked vehicle on the cement road, with a driving distance of 776.5 m. The red solid line in the figure represents the road section where the tracked vehicle predicted the ${\hat{T}}_{e}$ and estimated the $\hat{f}$. During the whole driving process of the tracked vehicle, the engine output torque was predicted and the rolling resistance coefficient was estimated four times. The total length of the driving section was 554.3 m. Figure 17 shows the speed, the displacement of the clutch cylinder, and the change in the gear when the tracked vehicle was driving on the cement road. The tracked vehicle did not shift again after starting in second gear, and it ran in second gear to complete the trip. The maximum speed was 14.36 km/h and the time was 260 s. The time for the tracked vehicle to meet the set driving conditions was 154.4 s. Figure 18 shows the changes in the acceleration, pitch angle, and heading angle of the tracked vehicle. The acceleration changed smoothly when the tracked vehicle was under the set working conditions, the pitch angle change was less than the set threshold, and the heading angle change was small. The vehicle could be considered to be in a straight driving state. Figure 19 shows the relationship between the engine speed and the percentage change in the accelerator pedal. The engine speed and the driver’s control accelerator pedal changed smoothly during the selected driving data.

Figure 20 shows the predicted values of the ${\hat{T}}_{e}$ and the estimated $\hat{f}$ values when the tracked vehicle ran on the cement road and met the set driving conditions for the first time. The ${\hat{T}}_{e}$ predicted by the engine output torque prediction model established by the GA–BP neural network was closer to the real value of the engine output torque. Figure 21 shows the estimation of the road rolling resistance coefficient $\hat{f}$ by the RLS algorithm. The GA–BP neural network was used to estimate the engine output torque and further estimated the road rolling resistance coefficient with better accuracy.

Table 3 shows the estimated values on the vehicle state, ${\hat{T}}_{e}$, and $\hat{f}$ of the tracked vehicle on the cement road under the set conditions. The $\sigma_{e}$ value in the results estimated by the GA–BP neural network was 13.09, and the value for the BP neural network method was 42.4% greater. The R from the GA–BP method was 0.895, which was 12.8% higher than that of the BP neural network. Figure 22 shows the change in the $e_{mb}$ values for the $\hat{f}$ estimation results obtained by the two methods. The $e_{mb}$ value by the GA–BP neural network predictions was 0.00061, which was 38.1% higher than that of the BP neural network predictions.

## 6. Conclusions

In this paper, a method for establishing an engine output torque prediction model based on vehicle driving data was proposed for tracked vehicles, and the model was used to further estimate the rolling resistance coefficient of the road. The following conclusions can be drawn from the experimental results:(1)The engine output torque prediction model obtained by fitting the vehicle driving data with the GA–BP neural network had a high level of engine output torque prediction accuracy. The engine output torque prediction model was established using vehicle driving data, which reduced the calibration work in the engine bench test stage significantly and had real-time updating capabilities. This method provides a new option for the establishment of an engine output torque model.(2)In this study, a prediction model of the engine output torque was established, and the RLS algorithm was used to estimate the road rolling resistance coefficients of tracked vehicles under certain driving conditions. The experimental results showed that when the tracked vehicle was driving on a sand road and a cement road, the rolling resistance coefficient of the road could be automatically estimated and had high accuracy when the vehicle driving state satisfied the set driving conditions. To a certain extent, this method meets the requirements for the real-time estimation of the rolling resistance coefficient of a road when a tracked vehicle drives longitudinally.(3)Limited by the system structure of the tracked vehicle and the measurement error of the sensor, to ensure the prediction accuracy of the engine output torque prediction model and the estimation accuracy of the road rolling resistance coefficient, it is necessary to limit the driving conditions of the tracked vehicle, which makes it difficult to apply this model throughout the whole driving process. Determining how to make the tracked vehicle estimate the road parameters over the whole driving process will be the focus of future research.

## Figures and Tables

**Figure 1 sensors-23-07549-f001:**
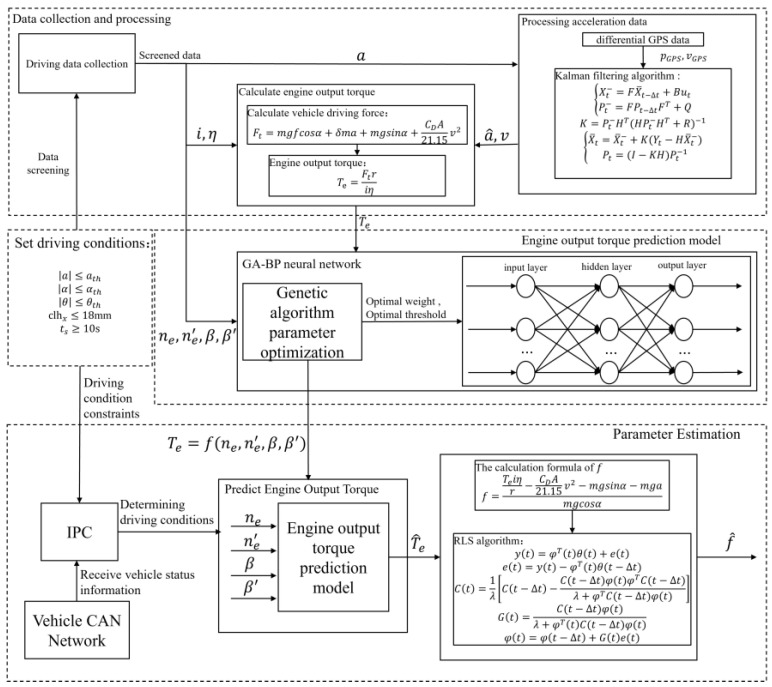
Process of engine output torque prediction and rolling resistance coefficient estimation.

**Figure 2 sensors-23-07549-f002:**
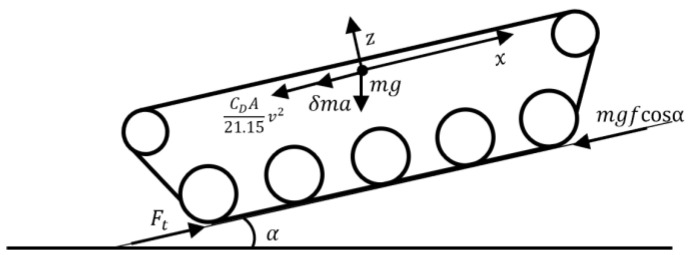
Forces on the tracked vehicle during longitudinal driving.

**Figure 3 sensors-23-07549-f003:**
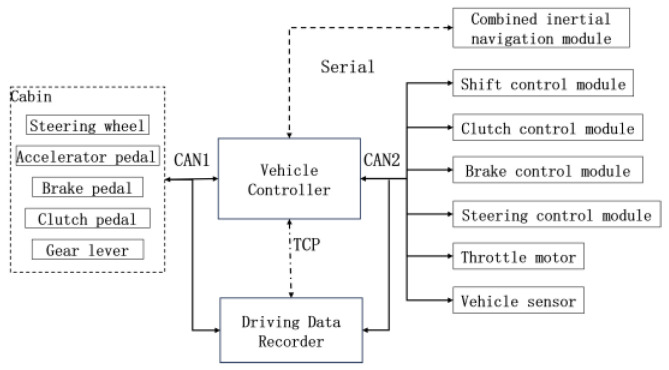
Communication structure of each module.

**Figure 4 sensors-23-07549-f004:**
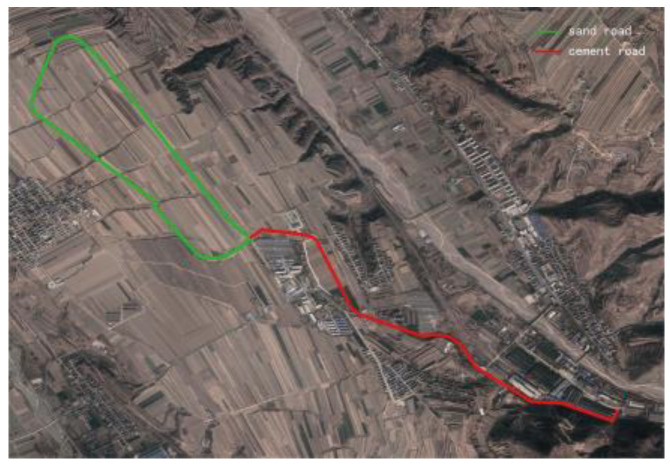
Satellite map of the experimental site.

**Figure 5 sensors-23-07549-f005:**
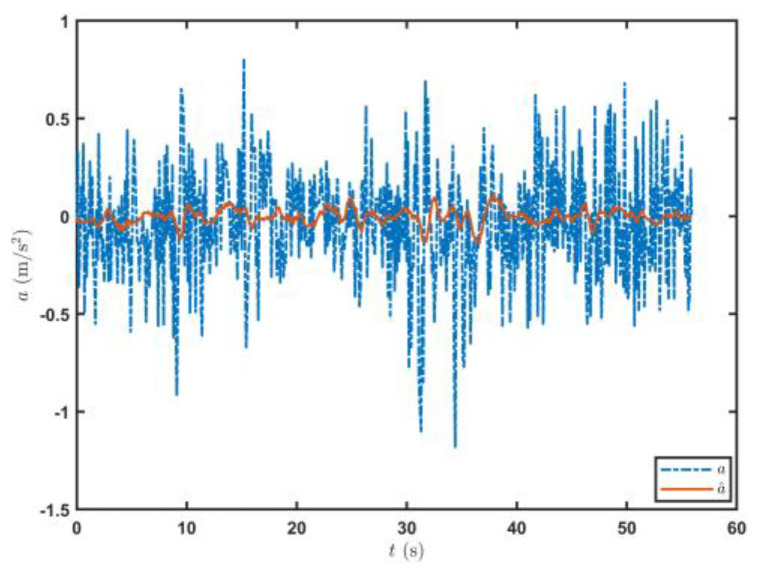
Filtering effect on the acceleration signal.

**Figure 6 sensors-23-07549-f006:**
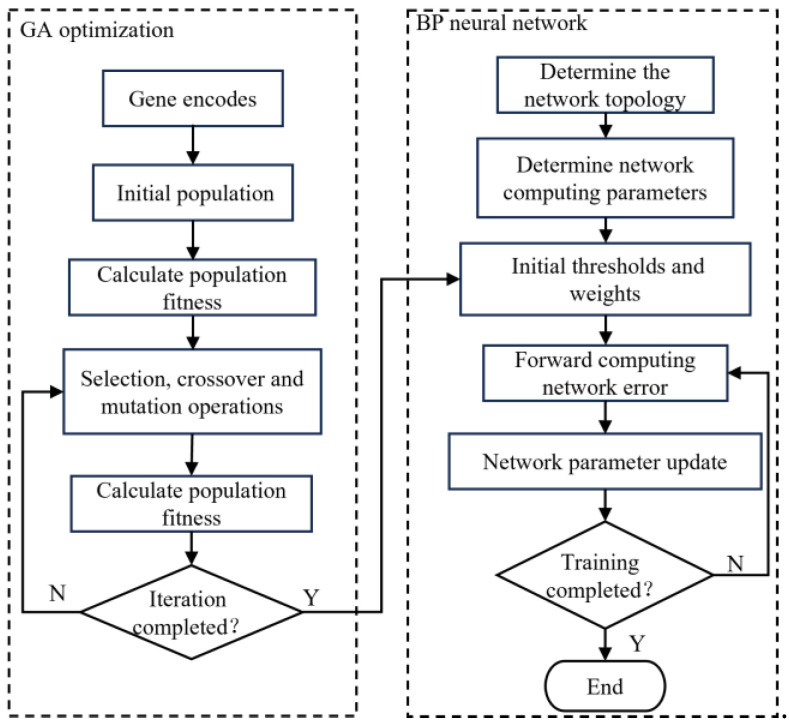
Genetic algorithm–backpropagation (GA) neural network algorithm structure diagram.

**Figure 7 sensors-23-07549-f007:**
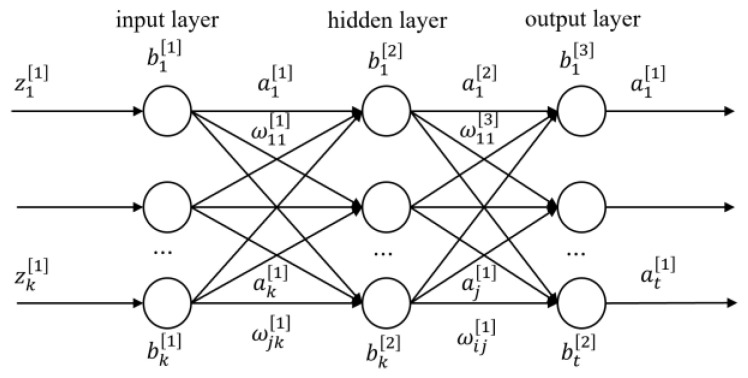
Structure of the backpropagation (BP) network.

**Figure 8 sensors-23-07549-f008:**
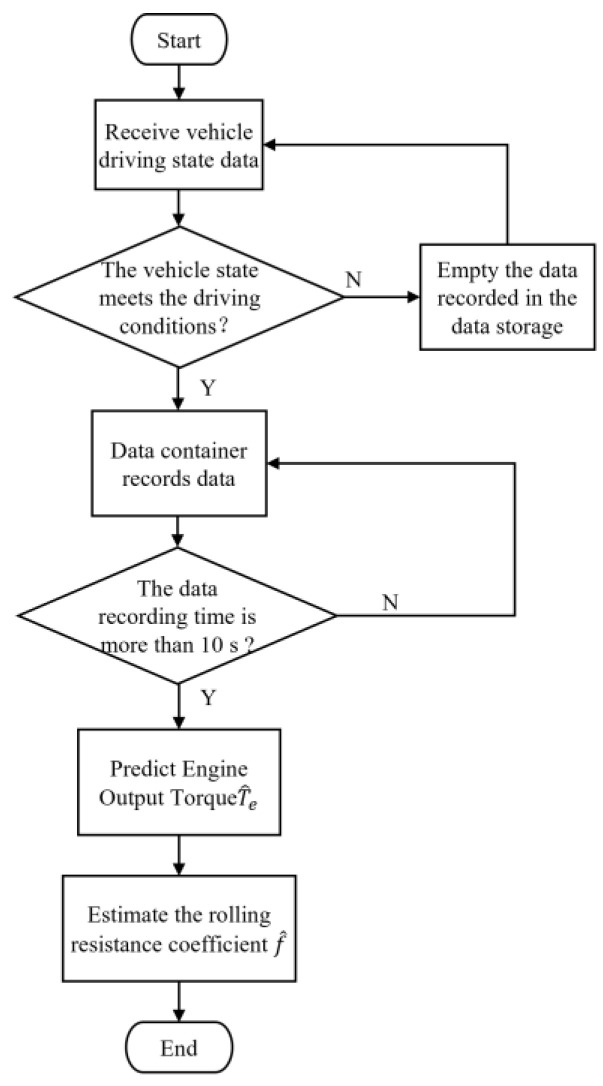
Process through which the industrial personal computer (IPC) processed the tracked vehicle driving data.

**Figure 9 sensors-23-07549-f009:**
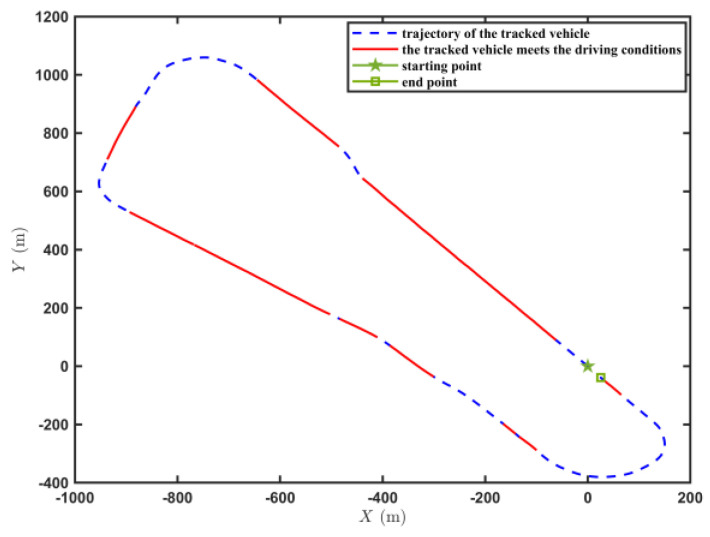
Trajectory of the tracked vehicle driving on the sand road.

**Figure 10 sensors-23-07549-f010:**
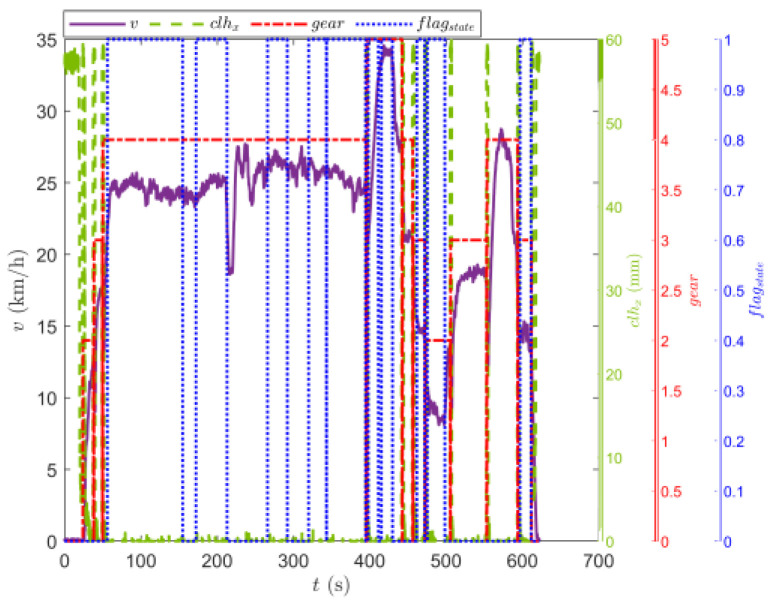
Changes in the speed, and gear and clutch cylinder displacement in the tracked vehicle on the sand road.

**Figure 11 sensors-23-07549-f011:**
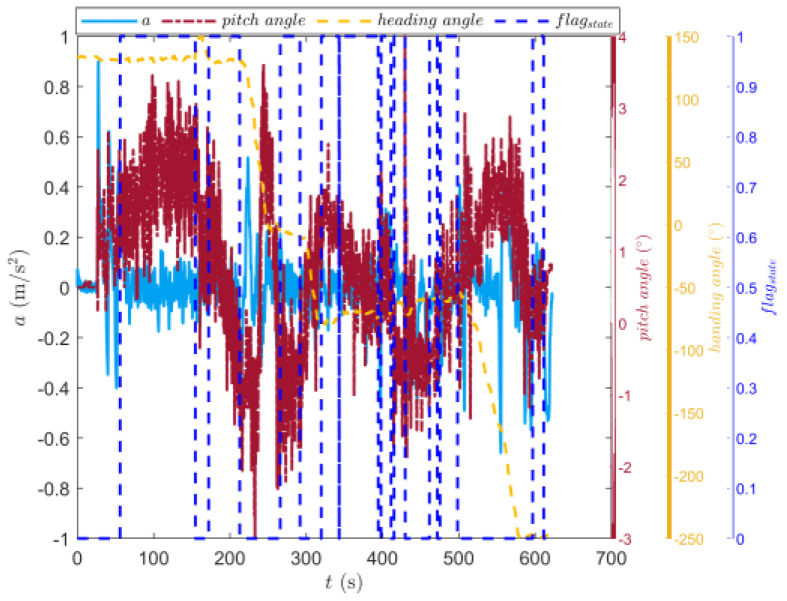
Changes in the longitudinal acceleration, heading angle, and pitch angle of the tracked vehicle running on the sand road.

**Figure 12 sensors-23-07549-f012:**
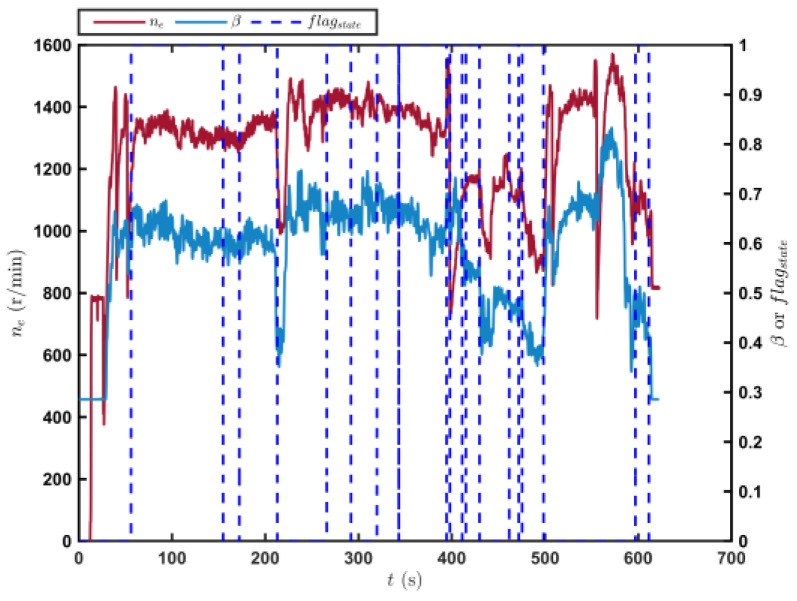
Changes in the engine speed and accelerator pedal position in the tracked vehicle running on the sand road.

**Figure 13 sensors-23-07549-f013:**
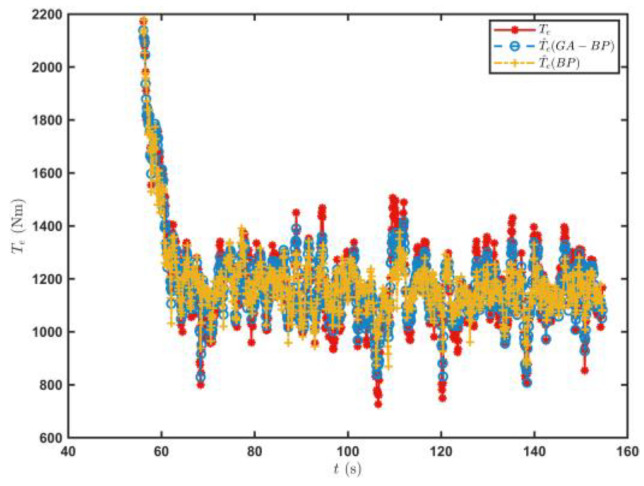
Predicted ${\hat{T}}_{e}$ values when the tracked vehicle satisfied the driving conditions for the first time on the sand road.

**Figure 14 sensors-23-07549-f014:**
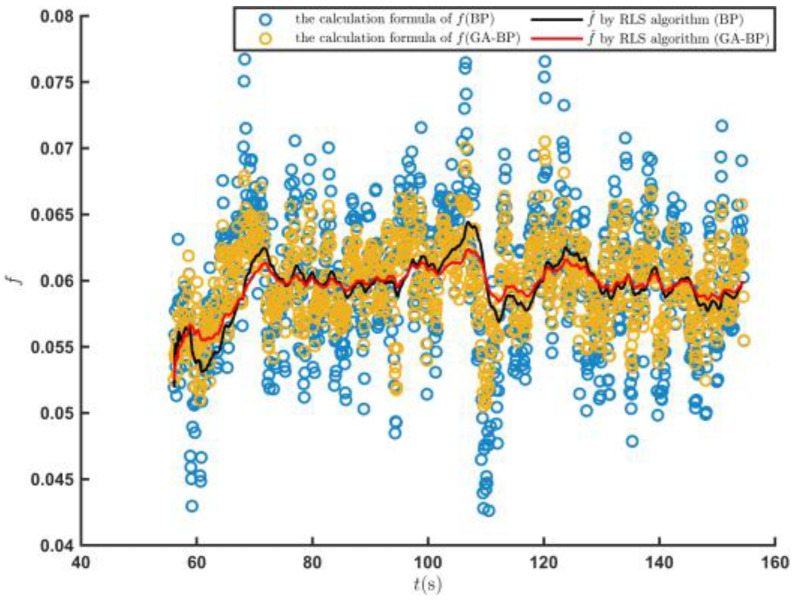
Predicted $\hat{f}$ values when the tracked vehicle satisfied the driving conditions for the first time on the sand road.

**Figure 15 sensors-23-07549-f015:**
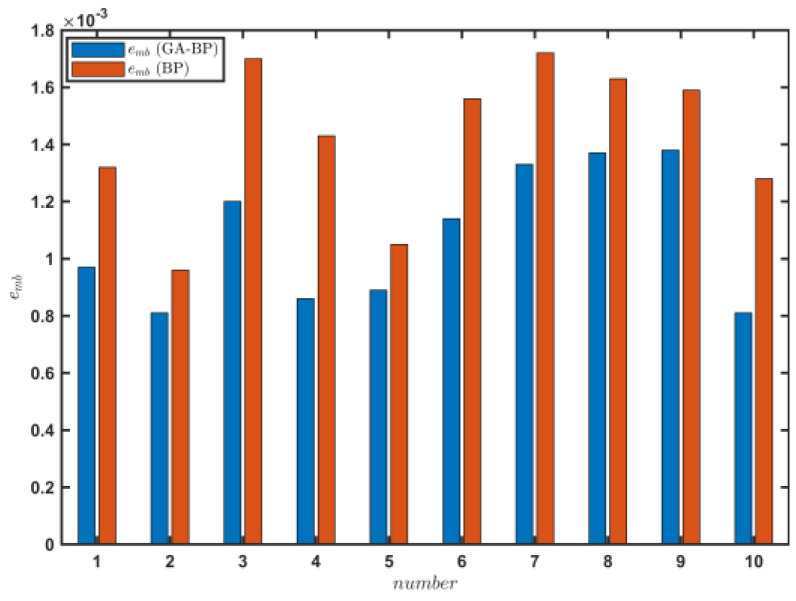
Engine output torque predictions when the tracked vehicle satisfied the driving conditions during 10 periods on the sand road.

**Figure 16 sensors-23-07549-f016:**
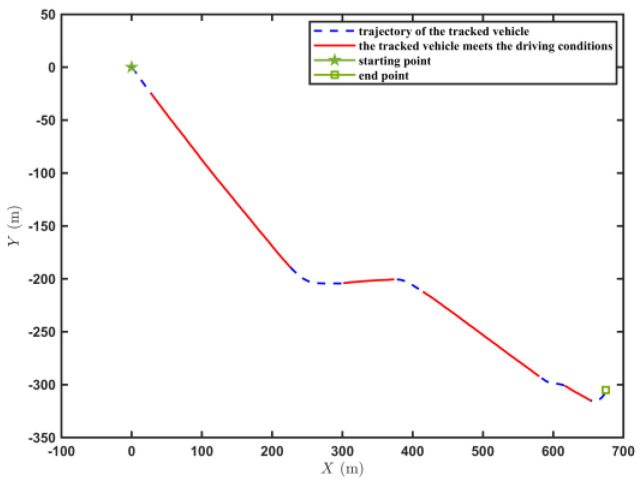
Trajectory of the tracked vehicle driving on the cement road.

**Figure 17 sensors-23-07549-f017:**
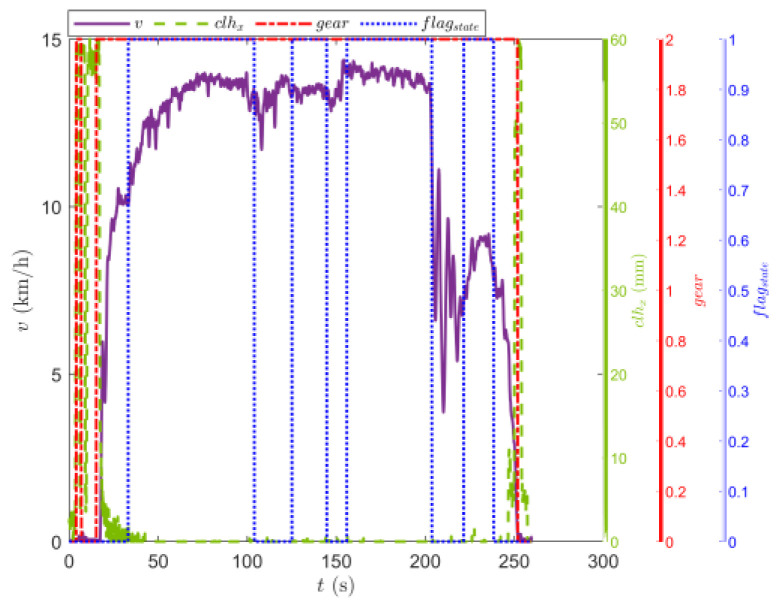
Changes in the speed, and gear and clutch cylinder displacement in the tracked vehicle on the cement road.

**Figure 18 sensors-23-07549-f018:**
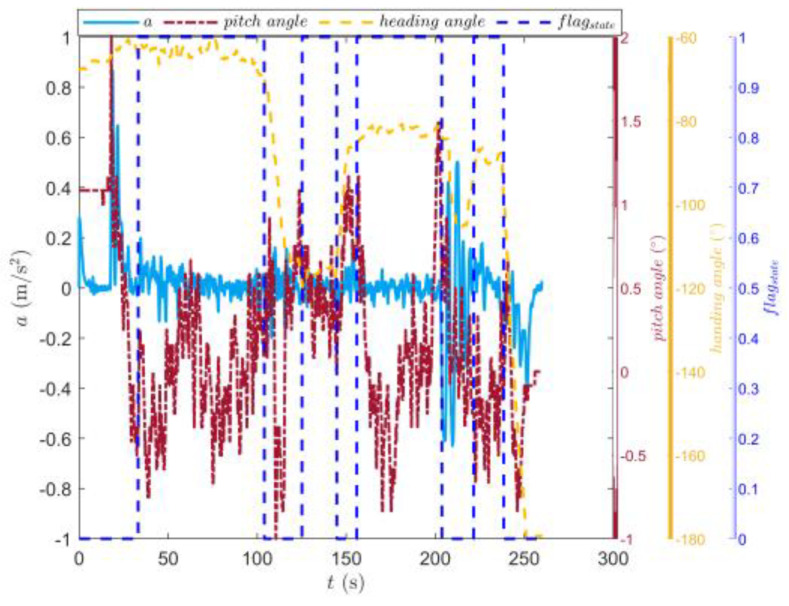
Changes in the longitudinal acceleration, heading, and angle of the tracked vehicle running on the cement road.

**Figure 19 sensors-23-07549-f019:**
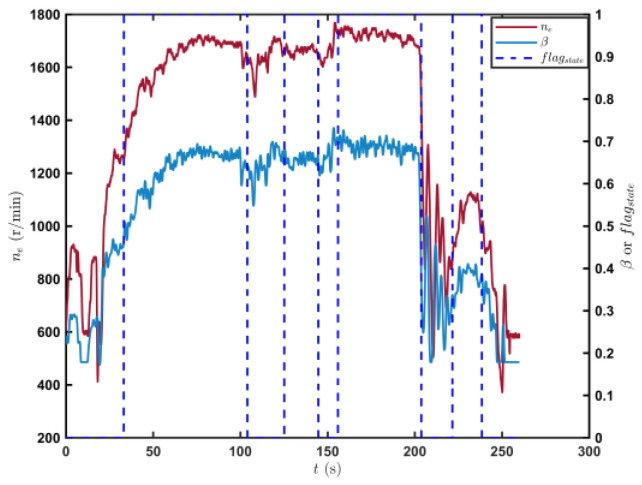
Changes in the engine speed and accelerator pedal position in the tracked vehicle running on the cement road.

**Figure 20 sensors-23-07549-f020:**
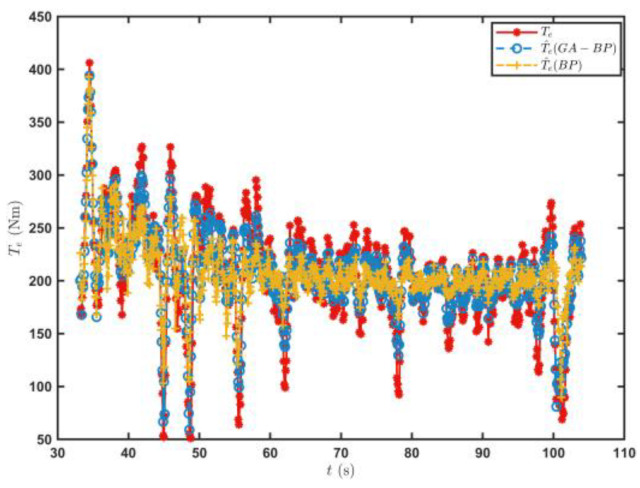
Prediction of the ${\hat{T}}_{e}$ values when the tracked vehicle satisfied the driving conditions for the first time on the cement road.

**Figure 21 sensors-23-07549-f021:**
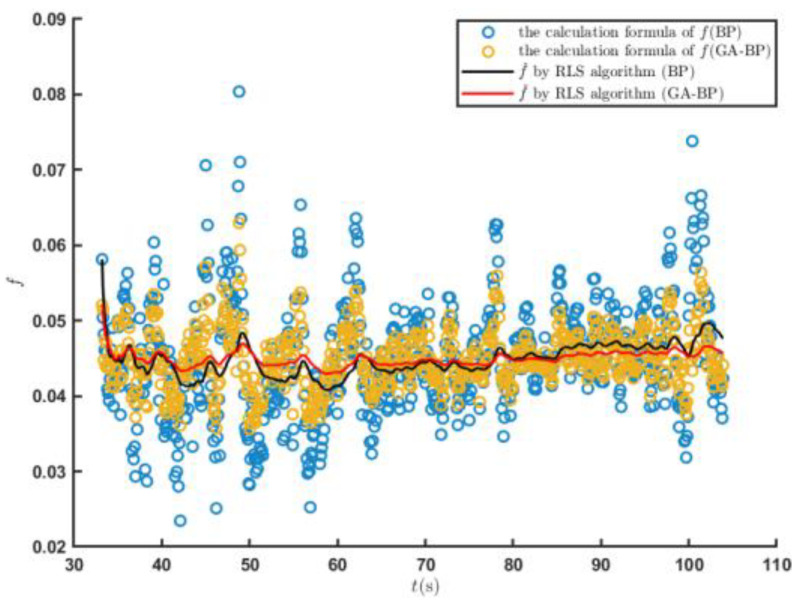
Prediction of the $\hat{f}$ values when the tracked vehicle satisfied the driving conditions for the first time on the cement road.

**Figure 22 sensors-23-07549-f022:**
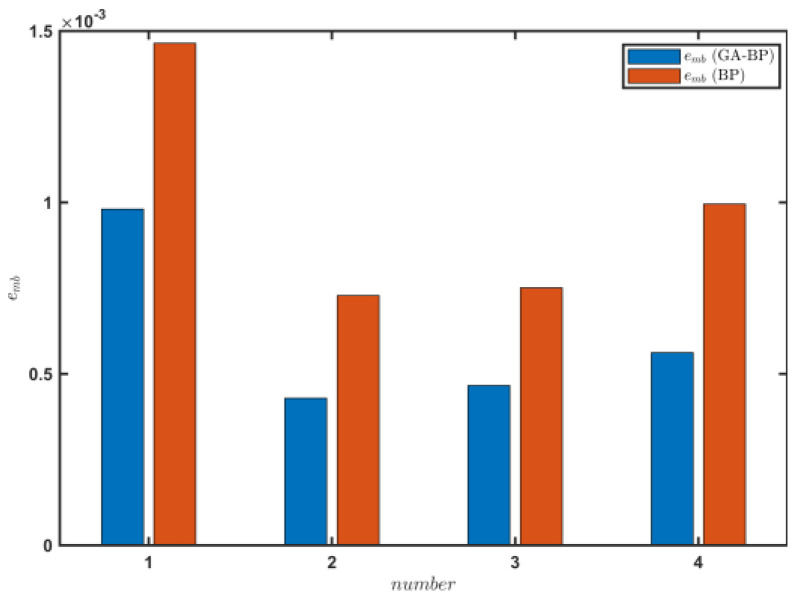
Engine output torque prediction when the tracked vehicle satisfied the driving conditions during 10 periods on the cement road.

**Table 1 sensors-23-07549-t001:** Tracked vehicle structural parameters.

Parameter	Value
m (kg)	31,000
r (m)	0.283
A ($m^{2}$)	6
$C_{D}$	0.45
δ	1.24
i (1st gear to 5th gear)	28.35/13.23/9.45/6.71/4.3
η (1st gear to 5th gear)	0.79/0.77/0.76/0.75/0.73

**Table 2 sensors-23-07549-t002:** Vehicle state, ${\hat{T}}_{e}$, and estimated $\hat{f}$ when the tracked vehicle satisfied the driving conditions on the sand road.

	Time (s)	Distance (m)	Gear	Average Speed (km/h)	$\sigma_{e}$ (GA–BP)	$\sigma_{e}$ (BP)	R (GA–BP)	R (BP)
1	56.1–154.6	671.13	4	24.48	43.6	79.69	0.9287	0.7624
2	172.2–212.8	279.84	4	24.8	32.34	60.07	0.9305	0.7604
3	266–291.8	189.63	4	26.46	43.47	75.6	0.8727	0.6452
4	319.8–343.1	167.53	4	25.88	35.83	75.93	0.903	0.5640
5	343.3–394.3	356.49	4	25.11	40.59	76.04	0.9118	0.6907
6	398.1–411.1	96.98	5	26.82	42.59	79.97	0.9498	0.7437
7	415.2–429.6	136.84	5	33.99	52.66	79.26	0.9387	0.7832
8	461.6–471.7	41.89	3	14.78	51.73	69.84	0.9184	0.7142
9	475.5–498.3	57.39	2	9.04	42.95	64.35	0.8942	0.7627
10	596.9–611.2	56.83	3	14.3	36.67	78.73	0.9215	0.7608

**Table 3 sensors-23-07549-t003:** Vehicle state, ${\hat{T}}_{e}$, and estimated $\hat{f}$ when the tracked vehicle satisfied the driving conditions on the cement road.

	Time (s)	Distance (m)	Gear	Average Speed (km/h)	$\sigma_{e}$ (GA–BP)	$\sigma_{e}$ (BP)	R (GA–BP)	R (BP)
1	33.3–104	257.9	2	13.12	19.01	33.38	0.86	0.75
2	125.3–144.7	75.5	2	13.45	11.2	18.12	0.83	0.748
3	156.1–203.7	184.1	2	13.89	10.49	18.54	0.92	0.76
4	221.7–238.4	39.8	2	8.51	11.66	20.88	0.89	0.81

## Data Availability

No new data was created.

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
