# Peer review of "Research on the Prediction Model of Engine Output Torque and Real-Time Estimation of the Road Rolling Resistance Coefficient in Tracked Vehicles"

_sensors, 2023, doi:10.3390/s23177549_

Round 1

Reviewer 1 Report

1. The introduction should be improved by addressing the main feature of the work and the motivation of the research. The contributions of the study should be given more clearly.

2. The statements in lines 151 and 152 are inconsistent. How to use Eq. (8) to calculate parameter () ?

3. Is the GA algorithm optimized online or offline? How does the author ensure computational efficiency?

4. In the design of the GA-BP and RLS, many parameters are used. It is better to explain the parameter value and how to select it?

5. What is the computational complexity of the proposed technique? The comparison between GA-BP and BP is insufficient, author must compare with the existing techniques.

6. The latest literature cited in the paper is from 2021, and the author needs to update the references. In addition, there are formatting errors in the references. Regarding the literature survey, the reviewer recommends to add a few more papers related to this study. Such as: En Lu, Wei Li, Song Jiang, et al. Anti-disturbance speed control of permanent magnet synchronous motor based on fractional order sliding mode load observer [J]. IEEE Access, 2022. DOI: 10.1109/ACCESS.2022.3214205.

7. Several English grammar errors must be carefully checked and corrected.

 Several English grammar errors must be carefully checked and corrected.

Author Response

Thank you very much for the advice of the experts. The suggestion is of great help to the improvement of manuscript quality. The contents of the manuscript are revised one by one according to the suggestions of experts. The specific content of the modification is described in detail in the description document. Please experts to view the document. If the experts have any questions, please timely, thank you very much.

Reviewer 2 Report

Research on Prediction Model of Engine Output Torque and Real-Time Estimation of Road Rolling Resistance Coefficient of Tracked Vehicles

Weijian Jia,Xixia Liu, Guodong Jia, Chuanqing Zhang, Bin Sun

The authors present a method for establishing an engine output torque prediction model based on vehicle driving data for tracked vehicles.

The topic addressed in this investigation is important, worthy investigation, and lies within the scope of this prestigious journal.

The reviewer’s remark

1)     In the introduction, the authors' should point the contribution to the field. The literature survey (17 item) is to poor for this field.

2)     What do the authors mean when they write “θ(t-1)? (Eq.5). In reviewer's opinion the authors should write in all equations θ(t- Δt) and define time step size Δt.

3)     Eqs 5 and 6 should be written in the notation adopted by the authors in Eqs 6-8.

4)     “d” in all Eqs non Italic! (see Eq.5)

5)     Authors should provide all data for calculations. There is no data for the neural network. What is the number of input and output layers? What were the authorities guided in their selection?

Author Response

(The authors gave the same response as above.)

Reviewer 3 Report

Review Report for

Research on Prediction Model of Engine Output Torque and Real-Time Estimation of Road Rolling Resistance Coefficient of Tracked Vehicles

This paper proposes a method for establishing a prediction model of the engine output torques of tracked vehicles based on vehicle driving data was proposed, and the road rolling resistance coefficient ? was further estimated using the model.

The article has clear logic and relatively complete content, but some shortcomings remain.

1.         Please provide comments on meeting the vehicle's driving status.

2.         Figures 12 and 19 lack a vertical axis.

3.         The symbols in the legend are missing annotations.

4.         The quality of the images in the article needs to be improved.

The language of the article needs to be improved to some extent.

Author Response

(The authors gave the same response as above.)
